# Rapid molecular diagnosis of Parechovirus infection using the reverse transcription loop-mediated isothermal amplification technique

**Tadafumi Yokoyama**[1]*, **Yuko Tasaki**[1], **Natsumi Inoue**[1], **Naotoshi Sugimoto**[1],
**Eri Nariai**[2], **Sanae Kuramoto**[2], **Taizo Wada**[1]

1 Department of Pediatrics, Kanazawa University, Kanazawa, Ishikawa, Japan, 2 Health and Food Safety
Department, Ishikawa Prefectural Institute of Public Health and Environmental Science, Kanazawa, Ishikawa,
Japan

* tadafumiy@staff.kanazawa-u.ac.jp

## Abstract

### Objectives

Human parechovirus (HPeV), especially HPeV A3 (HPeV3), causes sepsis-like diseases
and sudden infant death syndrome in neonates and young infants. Development of rapid
and easier diagnostic laboratory tests for HPeVs is desired.

### Methods

Original inner primers, outer primers, and loop-primers were designed on the 5′ untranslated
region of HPeV3. HPeV3 ribonucleic acids (RNAs), other viral RNAs, and clinical stool sam-
ples were used to confirm whether the designed primers would allow the detection of
HPeV3 with the reverse transcription loop-mediated isothermal amplification (RT-LAMP)
technique.

### Results

Three combinations of primers were created and it was confirmed that all primer sets
allowed the detection of HPeV3 RNAs. The primer sets had cross-reactivity with HPeV type
1 (HPeV1), but all sets showed negative results when applied to coxsackievirus, echovirus,
enterovirus, norovirus, and adenovirus genomes. Four of six stool samples, obtained from
newborn and infant patients with sepsis-like symptoms, showed positive results with our
RT-LAMP technique.

### Conclusions

This manuscript is the first description of an RT-LAMP for the diagnosis of HPeVs, allowing
a faster, easier, and cheaper diagnosis. This technique is clinically useful for newborns and
infants who have sepsis-like symptoms.

journal.pone.0260348

SAUDI ARABIA

**Data Availability Statement:** The sequence of
HPeV3 is held in GenBank of the National Center

for Biotechnology Information (Bethesda, MD, USA). Accession number is AB084913.1.

**Funding:** This project was supported by the Morinaga Foundation for Health & Nutrition (http://www.disclo-koeki.org/02a/00327/index.html). The funders had no role in the conceptualization, analysis, interpretation, or decision to publish this manuscript.

**Competing interests:** The authors have declared that no competing interests exist.

## Introduction

Human parechovirus (HPeV) is increasingly being recognized as a potentially severe viral infection in neonates and young infants [1–3]. HPeVs usually cause the "common cold" (i.e., respiratory or gastrointestinal illness similar to rhinovirus, enterovirus, and coronavirus) [1–3]. However, HPeVs have been implicated in cases of neonatal sepsis-like diseases, encephalitis, sudden infant death syndrome, and myocarditis [1–3]. HPeVs belong to the family *Picornaviridae* and are divided into 19 genotypes [1].

Human parechovirus A3 (HPeV3) is the most clinically important genotype among HPeVs because of its association with severe disease in newborns and young infants [1, 3]. HPeV3-infected infants can present with a sepsis-like symptom, often with central nervous system involvement, which is difficult to differentiate clinically from bacterial sepsis [1, 3]. They may present with fever, tachycardia, and erythema [1]. Abdominal distension and navel protrusion are also seen in these patients [1]. Characteristic laboratory findings are cytopenia and elevated lactate dehydrogenase and ferritin levels, which suggest hemophagocytic lymphohistiocytosis [1]. Seizures or significant neurological impairment sometimes occur [1]. Severe HPeV3 infection in infants is also associated with a risk of long-term complications [1].

The application of molecular diagnostic methods enables the early recognition of HPeV infections. Early recognition is important because it may reduce the use of antibiotics and shorten the duration of hospital admissions for patients with mild to moderate disease. Early diagnosis is also likely to lead to appropriate investigations and follow-up for potential complications in infants who are severely affected. In some research institutes, HPeVs are detected by using nested polymerase chain reaction (PCR) and direct sequencing of amplified PCR products [1–3]. Real-time PCR is also widely used. However, it would be useful if there was a faster, easier, and cheaper diagnostic test for HPeVs.

We developed an easy and faster molecular diagnostic method to detect HPeV by using the reverse transcription loop-mediated isothermal amplification (RT-LAMP) technique. Using the RT-LAMP technique presented in this study, it was possible to diagnose HPeV in 2 h in small steps.

## Materials and methods

### Design of the RT-LAMP primers

The sequence of HPeV3 (GenBank: AB084913.1) was downloaded from the National Center for Biotechnology Information (Bethesda, MD, USA). Based on the highly conserved 5′ untranslated region (UTR) sequence of HPeV3, inner primers (i.e., FIP and BIP), outer primers (i.e., F3 and B3), and loop-primers (loop-F and loop-R) were originally designed on the 5′ UTR of HPeV3 by using PrimerExplorer V5 (Eiken Chemical Co., Ltd., Taito-ku, Tokyo, Japan).

### RT-LAMP assay

HPeV3 viral ribonucleic acid (RNA), used as a positive control, was kindly provided by the Health and Food Safety Department at Ishikawa Prefectural Institute of Public Health and Environmental Science (Kanazawa, Ishikawa, Japan). For the detection of viral RNA by LAMP assay, the Loopamp RNA Amplification Kit (Eiken Chemical Co., Ltd., Taito-ku, Tokyo, Japan) was used following the manufacturer's protocol. Viral RNA was incubated with enzymes, buffers, dNTPs, and primers at 63°C for 1 h. Then, the enzymes were deactivated at 80°C for 5 min. The products obtained by RT-LAMP were visually detected. The RT-LAMP

products were confirmed by direct sequencing using other originally designed primers (S1 Table).

## Direct sequencing of HPeV gene amplified with nested PCR and RT-LAMP

Complementary DNA (cDNA) was synthesized using Superscript II Reverse Transcriptase (Invitrogen, Carlsbad, CA, USA). Stool samples were first screened with nested PCR targeting the partial 5′ UTR of HPeVs. The samples that tested positive were assayed using nested PCR targeting the VP1 region for genotyping [2, 3]. Amplified DNA was directly sequenced using the BigDye™ Terminator v3.1 Cycle Sequencing Kit (Thermo Fisher Scientific, Waltham, MA, USA) in both directions. Each genotype was determined by comparing the nucleotide sequence with the available HPeV sequences from GenBank using the Genetyx program (GENETYX, Shibuya-ku, Tokyo, Japan). For RT-LAMP products, as for nested PCR, the amplified DNA was directly sequenced using the BigDye Terminator cycle sequencing kit (Thermo Fisher Scientific, Waltham, MA, USA) in both directions by using primers for sequencing (S1 Table).

## Sensitivity and specificity analysis of HPeV RT-LAMP

Virus-infected cells (Vero/E6 cells supplied from the National Institute of Infectious Diseases in Japan (https://www.niid.go.jp/niid/ja/)) were cultured and two more HPeV3 were harvested, one HPeV1, one adenovirus serotype 1, one coxsackievirus B5, one echovirus 11, one enterovirus D68, and one enterovirus A71. The samples were collected through a medically prescribed test and completely de-identified before the samples were accessed.

Additionally, the stools of one patient with adenovirus gastroenteritis and three patients with norovirus gastroenteritis were collected after obtaining written informed consent from the parents. Adenovirus DNA and norovirus RNA were extracted from these stools. The patients were diagnosed by rapid diagnostic tests.

All samples (applied viral nucleic acid amount is 62–296 ng/sample) were tested for HPeVs by using RT-LAMP.

## RT-LAMP analysis using clinical samples

The stools of six anonymous febrile infants were obtained at the time of the suspicion of HPeV infection, after obtaining written informed consent from the parents. Stool samples of 75 to 150 mg were diluted with 200 μL of distilled water and vortexed strongly for 5 min. The samples were then centrifuged at 20,000*g* for 5 min at 4˚C. For the extraction of viral RNA, 150 μL of the supernatants were applied. Viral RNA was extracted from supernatants by using the QIAmp Viral RNA Mini Kit (Qiagen, Hilden, Germany), following the manufacturer's instructions. Extracted viral RNA was stored at −80˚C until amplification using RT-LAMP and nested PCR detection.

## Ethics

This study was approved by the institutional review board of the Graduate School of Medical Sciences at Kanazawa University (Kanazawa, Japan; protocol number: 2014-072(1686)). This study was conducted according to the Declaration of Helsinki. All experiments were performed according to relevant guidelines and regulations (including informed consent from all participants and parents).

## Results

### 1. Development of RT-LAMP method for HPeVs

The 5′ UTR is 700 bp in length. While designing primers using the whole 5′ UTR region via PrimerExplorer V5 (Eiken Chemical Co., Ltd., Taito-ku, Tokyo, Japan), no appropriate combinations of primers were found. Next, by designing primers using a part of the 5′ UTR region 5′ UTR (e.g., 1–400, 11–410, 21–340, . . ., 301–700 (i.e., the last 5′ UTR), a total of 42 primer combinations (i.e., FIP, BIP, F3, and B3) were found. The results showed eight candidate combinations (S2 Table). For example, using the 61–460 sequence of the 5′UTR, 87 F1 sequences, 90 F2 sequences, . . ., and 115 BIP sequences were found as candidate combinations. However, there was only one optimal combination for the F1, F2 to BIP LAMP method. When the primer was created using the 5′UTR 71–470, two optimal combinations were found. In this way, the primer combination was comprehensively searched while shifting the 400 bp sequence by 10 bp.

From these 42 combinations, 3 combinations (Set A$^{loop\text{-}}$, B$^{loop\text{-}}$, and C$^{loop\text{-}}$) were most frequently duplicated (12 times), but the other five primer combinations were extracted only once or twice (Fig 1A and Table 1).

When HPeV3 viral RNAs were used for RT-LAMP, Sets A$^{loop\text{-}}$, B$^{loop\text{-}}$ and C$^{loop\text{-}}$ showed positive results. However, after testing multiple times, these sets had unstable or unreproducible rates (positive rate, %; (i.e., number of positive samples/number of tested samples)), as follows: Set A$^{loop\text{-}}$, 33.3% (6/18); Set B$^{loop\text{-}}$, 44.4% (8/18); and Set C$^{loop\text{-}}$, 86.0% (6/7).

Therefore, two more primers were added: loop-F and loop-R, to increase the sensitivity of the RT-LAMP method. Loop-R was automatically designed by using PrimerExplorer V5 (Eiken Chemical Co., Ltd.), but loop-F was manually designed because PrimerExplorer V5 could not answer the loop-F candidate (Fig 1A and Table 1). As a result, we could detect HPeV3 by using primer Set A (Set A$^{loop\text{-}}$ + loop-F/R), Set B (Set B$^{loop\text{-}}$ + loop-F/R), and Set C (Set C$^{loop\text{-}}$ + loop-F/R) (Fig 1B). The RT-LAMP products that included the target sequence and the reverse complementary sequence were confirmed by direct sequencing.

### 2. Minimum reaction time and amount of viral RNAs

The RT-LAMP was tested by changing the reaction times to 20 min, 30 min, 40 min, up to 110 min. The RT-LAMP method was also tested by applying $1.18 \times 10^2$ ng, $1.18 \times 10$ ng, $1.18$ ng, up to $1.18 \times 10^{-5}$ ng of total viral RNAs. Results are shown in the S3 Table. When more than $1.18 \times 10^{-1}$ ng of the total viral RNAs was applied, positive results were visible at 20 min in Set A and Set B. Set C showed optically positive results at 30 min. However, when less than $1.18 \times 10^{-1}$ ng of the total viral RNAs was applied, the reaction became unstable. Only Set C showed positive results at 50 min for all amounts of viral RNAs. Therefore, we confirmed that our RT-LAMP technique could detect at least 118 pg of the total viral RNAs in stool samples and could show positive results in 30 min.

The reaction temperature was also investigated, using 59˚C, 61˚C, 63˚C, 65˚C, and 67˚C. The most appropriate and stable temperature was 63˚C (S4 Table).

### 3. Specificity of the RT-LAMP method

RT-LAMP was conducted using samples of which the etiology was already known (two more HPeV3, one HPeV1, one coxsackievirus B5, one echovirus 11, one enterovirus D68, one enterovirus A71, three noroviruses, two adenoviruses. HPeV3 and HPeV1 showed positive results in our RT-LAMP assay. However, all other samples were negative (Fig 2 and S5 Table).

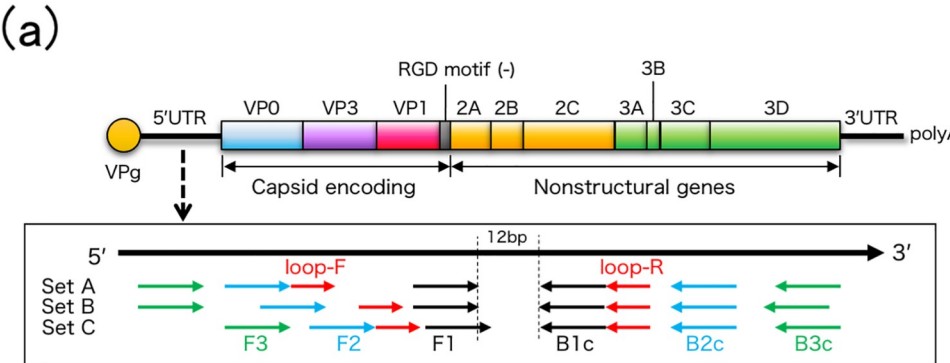

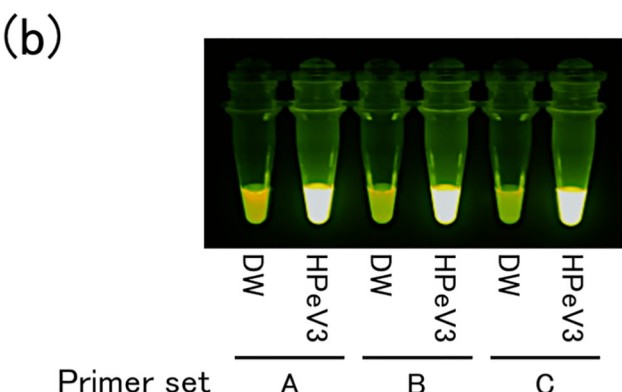

**Fig 1. Development of the RT-LAMP method for HPeVs.** (a) The HPeV3 genome and scheme of the combinations of the primers for HPeVs. Three primer sets for RT-LAMP were designed on the 5′ UTR of HPeV3. (b) Results of RT-LAMP using the three primer sets. Tubes containing HPeV3 RNAs show a bright fluorescence, compared with tubes with distilled water. DW, distilled water; HPeV, human parechovirus; HPeV3, human parechovirus A3; RT-LAMP, reverse transcription-loop-mediated isothermal amplification; UTR, untranslated region.

## 4. RT-LAMP analysis using clinical samples

The stools of six anonymous febrile infants were obtained at the time of suspected HPeV infection. From these stools, viral RNAs were extracted. The concentration of viral RNA solutions was 84.8 ± 25.2 μg/mL (expressed as the mean ± standard deviation).

Two microliters of viral RNA solution, containing 169.7 ± 50.4 ng of viral RNAs, was applied, for the RT-LAMP experiments. The LAMP products were visible in the stool samples of four patients (Patients #3–#6) and the positive control (Fig 3). Nevertheless, no RT-LAMP products were visible in the negative control and the samples of Patients #1 and #2. RT-LAMP experiments were conducted using both Sets A, B, and C, and all showed the same results (S1 Fig).

In the six stools, we simultaneously tried to detect HPeVs genomes by using the nested PCR technique. HPeV3 were detected in four samples and were not detected HPeVs in two samples. This finding corresponded to the RT-LAMP results. An important finding was that the results were known in 2 h after collecting stool samples when using the RT-LAMP technique, but in 2 days when using the nested PCR technique. In addition, all RT-LAMP products were confirmed by direct sequencing.

**Table 1. Primer information for RT-LAMP of HpeVs.**

| Set | Name | Sequence (5'→3') |
|---|---|---|
| A | F3 | **GATGGCGTGCCATAACTCT** |
| | FIP | GGTTCCCACACGTCATCAGACA–*GATACCACGCTTGTGGACC* |
| | B3 | **GAACCAATCCCAAAGGGTCT** |
| | BIP | CAGTTTGCTGCAAAGCATCCCA–*CTTGGCTTTTGGCCCCAG* |
| | loop-F | AGGATGGCTGTGTGAGCATAA |
| | loop-R | CTGCCAGCGGATCTACATCT |
| B | F3 | **GATGGCGTGCCATAACTCT** |
| | FIP | GGTTCCCACACGTCATCAGACA–*TGTGGACCTTATGCTCACAC* |
| | B3 | **CCAATCCCAAAGGGTCTGTT** |
| | BIP | CAGTTTGCTGCAAAGCATCCCA–*CTTGGCTTTTGGCCCCAG* |
| | loop-F | TTACAAACTTACTAGAGGATG |
| | loop-R | CTGCCAGCGGATCTACATCT |
| C | F3 | **GATACCACGCTTGTGGACC** |
| | FIP | AACAGGTTCCCACACGTCATCA–*ATGCTCACACAGCCATCC* |
| | B3 | **GAACCAATCCCAAAGGGTCT** |
| | BIP | CAGTTTGCTGCAAAGCATCCCA–*CTTGGCTTTTGGCCCCAG* |
| | loop-F | ACATCTTACAAACTTACTAG |
| | loop-R | CTGCCAGCGGATCTACATCT |

F1/B1c: Arial, *F2/B2c*: *Italic*, **F3/B3c: Bold**, loop-F/R: Times New Roman.

## Discussion

This study is the first to describe the detection of HPeVs by using the RT-LAMP method.

To diagnose HPeV infection, the PCR method is available. This method requires several steps, which take time to accomplish. This disadvantage of PCR tests is problematic for newborns and infants whose physical condition varies hourly. The real-time PCR method is also widely used for the current HPeV3 detection. However, the real-time PCR test requires specialized expertise and is available at limited clinical laboratories. Additionally, it is costly.

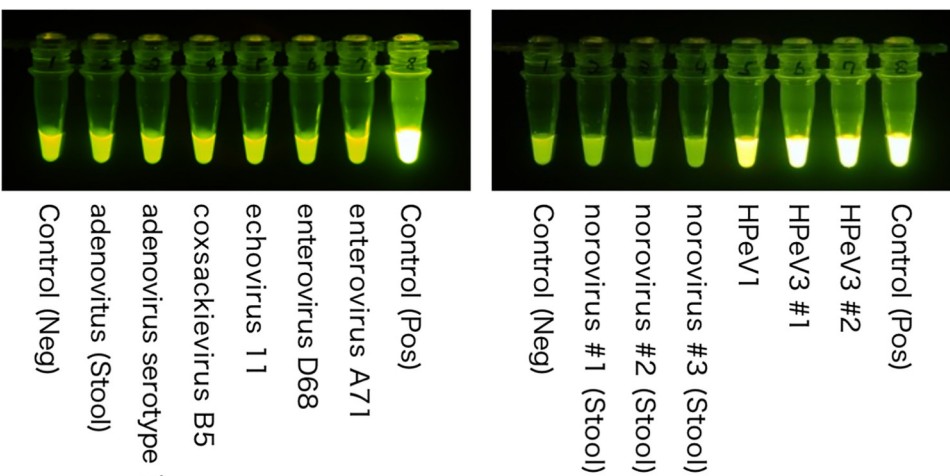

**Fig 2. Results of RT-LAMP using different viral genomes.** Tubes containing HPeV1 and HPeV3 show positive results, whereas the other tubes show negative results. HPeV1, human parechovirus type 1; HPeV3, human parechovirus type 3; Neg, negative; Pos, positive; RT-LAMP, reverse transcription loop-mediated isothermal amplification.

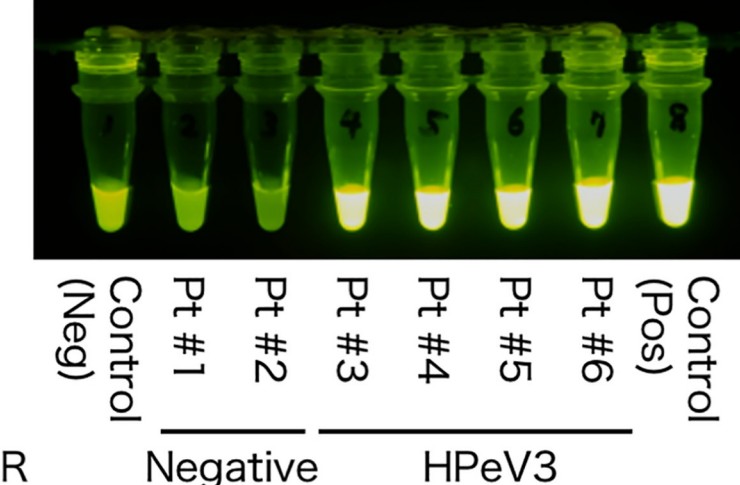

**Fig 3. Representative picture of the results of RT-LAMP by using clinical samples.** The stool samples of two patients (Patient #1 and Patient #2) show negative results. The stool samples of four patients (Patients #3–#6) show positive results for RT-LAMP. This figure is from primer Set A. The results of nested PCR for HPeV3 corresponded completely to the results of RT-LAMP. HPeV3, human parechovirus A3; Neg, negative; PCR, polymerase chain reaction; Pos, positive; RT-LAMP, reverse transcription loop-mediated isothermal amplification.

Therefore, we believe that a faster, easier, and cheaper diagnostic system to detect HPeVs may be clinically useful for these patients.

The LAMP reaction, an alternative nucleic acid amplification method developed by Notomi et al. [4], is based on strand displacement by DNA polymerase under isothermal conditions in which the temperature range is 60°C–65°C [4–7]. The LAMP assay is very specific, compared with other molecular detection methods, because four primers are necessary that recognize six specific regions of the target gene for amplification [4–7]. The method generates a large amount of amplification products in positive samples ($10^9$– to $10^{10}$–fold in 15–60 min), thereby allowing assessment of these products with the naked eye and fluorescent dye [4–7]. Another advantage of the LAMP method is that amplification occurs at one temperature; therefore, it does not require a thermal cycler [4–7]. The RT-LAMP assay, which is carried out with an RT step and a LAMP reaction in a single tube, is a simple, highly specific, rapid, and cost-effective method [6, 7]. The RT-LAMP method needs only a few operating steps, which is also advantageous in preventing contamination, and thus being safe for technicians.

In this study, we have developed an RT-LAMP assay for HPeV detection by designing specific primers, based on the 5′ UTR region of the HPeV3 viral gene. We compared *in silico* the primer alignment to the 5′ UTR region of other types of HPeVs by using the Basic Local Alignment Search Tool (National Center for Biotechnology Information, Bethesda, MD, USA). As a result, our primers showed some homology in HPeV1, HPeV4, HPeV5, HPeV8, HPeV14, HPeV 17, and HPeV18. Therefore, our RT-LAMP assay may have cross-reactivity to the other HPeVs, except for HPeV3. We indeed confirmed that HPeV1 was also detected in our primer sets.

Aside from HPeV3, HPeV infections are common around the world and have been identified on every inhabited continent [2]. However, an accurate prevalence rate is unknown and varies, depending on the age of the individuals included in the study population and the sampling sites chosen [2]. HPeV A1 (HPeV1) is the most prevalent genotype and most commonly causes gastrointestinal and respiratory diseases [1, 2]. The symptoms are usually mild, although HPeV1 sometimes causes respiratory disease in young children, which initially presents as coryza, accompanied by cough and dyspnea. Occasional HPeV1 cases present with

necrotizing enterocolitis, encephalitis, and Reye's syndrome [1, 2]. HPeV1 outbreaks some-times occur in nursery schools and neonatal intensive care units [1, 2]. The frequency of HPeV1 infection is not uncommon but unknown because clinicians cannot diagnose HPeV infection, including HPeV1 and HPeV3, as easily as with a rapid diagnosis kit (i.e., influenza virus). Clinicians may diagnose a patient as having "some kind of viral infection." In some more severe cases, a research institute can help to diagnose the infection.

Therefore, we believe that our RT-LAMP method withstands criticism because the most important purpose is the detection of HPeVs, regardless of the genotype of HPeVs as a screen-ing. However, further experiments are necessary to design HPeV type-specific primers and develop the appropriate LAMP conditions.

We completely detected HPeV3 from clinical stool samples. These patients were all 1-month-old babies and showed sepsis-like symptoms. These patients were successfully diag-nosed as having HPeV infection, in 2 h. Meanwhile, if the nested PCR technique had been used, it would have taken 2 days before the results were known. This delay in determining a diagnosis can be critical for infants and newborns. Our RT-LAMP technique may solve this problem.

Additionally, HPeV3 infection is usually asymptomatic or manifests as a faint common cold in older children and adults. HPeV3 is known to cause myalgia/myositis in patients of all ages and is recognized as an important pathogen for all ages [1]. Our RT-LAMP method may also be useful for myalgia/myositis in all ages.

In our experiment, we clarified that 50–100 μg/mL of total viral RNAs can be collected from 75 mg to 150 mg of stool. It was also confirmed that 100 pg to 500 ng of total viral RNAs were sufficient to conduct our RT-LAMP technique and obtain accurate results. Usually, 2–5 μL of viral RNA solutions was used; hence, 100–200 ng of total viral RNA. Thus, 75–150 mg of stool is sufficient for our LAMP method.

On account of the limited number of cases, we were unable to determine (1) the sensitivity and specificity of each primer set and (2) the usefulness of RT-LAMP, if applied to serum, urine, and cerebrospinal fluids samples. (3) the limit of detection determined using Probit analysis. To confirm these limitations, a large prospective study is necessary. Additionally, although we believe our LAMP method is useful for screening, it is necessary to examine whether real-time PCR or the LAMP method is faster and cheaper.

In conclusion, by using our RT-LAMP technique for HPeVs, an HPeV infection can be diagnosed faster, easier, and at low costs. It is clinically useful for all patients with gastrointesti-nal and respiratory diseases, myalgia/myositis, encephalitis, etc., especially for newborns and infants who have sepsis-like symptoms.

## Supporting information

**S1 Fig. The results of RT-LAMP using different primer sets.** The RT-LAMP experiments were conducted using Sets A, B, and C. All sets show the same results. Neg, negative; Pos, posi-tive; RT-LAMP, reverse transcription loop-mediated isothermal amplification.
(TIF)

**S1 Table. Primer information for direct sequencing.**
(PDF)

**S2 Table. List of candidates of the combinations of primers.**
(PDF)

**S3 Table. Results of the RT-LAMP for different RNA amounts and reaction times.**
(PDF)

**S4 Table. Result of the RT-LAMP changing reaction temperature.**
(PDF)

**S5 Table. Results of the RT-LAMP for different viruses.**
(PDF)

## Author Contributions

**Conceptualization:** Tadafumi Yokoyama, Naotoshi Sugimoto, Taizo Wada.

**Data curation:** Tadafumi Yokoyama, Yuko Tasaki, Natsumi Inoue, Naotoshi Sugimoto, Eri Nariai.

**Formal analysis:** Tadafumi Yokoyama, Yuko Tasaki, Natsumi Inoue, Naotoshi Sugimoto, Sanae Kuramoto.

**Investigation:** Tadafumi Yokoyama, Yuko Tasaki, Natsumi Inoue, Naotoshi Sugimoto, Eri Nariai.

**Methodology:** Tadafumi Yokoyama, Naotoshi Sugimoto.

**Project administration:** Taizo Wada.

**Resources:** Eri Nariai, Sanae Kuramoto.

**Supervision:** Eri Nariai, Sanae Kuramoto, Taizo Wada.

**Validation:** Tadafumi Yokoyama, Eri Nariai, Sanae Kuramoto.

**Visualization:** Tadafumi Yokoyama.

**Writing – original draft:** Tadafumi Yokoyama.

**Writing – review & editing:** Taizo Wada.

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
