## [Decision Letter · Decision Letter 0]

5 Aug 2021

PONE-D-21-06788

Rapid molecular diagnosis of Parechovirus infection using the reverse transcription loop-mediated isothermal amplification technique

PLOS ONE

Dear Dr. Yokoyama,

Thank you for submitting your manuscript to PLOS ONE. After careful consideration, we feel that it has merit but does not fully meet PLOS ONE’s publication criteria as it currently stands. Therefore, we invite you to submit a revised version of the manuscript that addresses the points raised during the review process.

We look forward to receiving your revised manuscript.

Kind regards,

Ahmed S. Abdel-Moneim, Ph.D.

Academic Editor

PLOS ONE

Journal Requirements:

2. Thank you for including your ethics statement:  "This study was approved by the institutional review board of Graduate School of Medical Sciences at Kanazawa University [Kanazawa, Japan; protocol number: 2014-072(1686)]. This study was conducted according to the Declaration of Helsinki. All experiments were performed in accordance with relevant guidelines and regulations (including informed consent from all participants and parents). We obtained informed consent from the parents when we collected the stools of anonymous febrile infants.".   

Please provide additional details regarding participant consent. In the ethics statement in the Methods and online submission information, please ensure that you have specified what type you obtained (for instance, written or verbal, and if verbal, how it was documented and witnessed). 

3. Please provide the data or graph for the temperature optimization experiment (line 174-175) as a supplementary file.

4. Please provide: the name(s) and source(s) of the cultured cells used for virus-infection, as well as a brief description of viral infection methods. If the samples were from a clinical source, please state whether the samples were:

(1) from an established biobank (if so please provide the name and a link)

(2) specifically collected for this study or not

(3) whether the samples were collected through a medically prescribed test

(4) whether the samples were completely de-identified before researchers accessed the samples

Reviewers' comments:

Reviewer's Responses to Questions

**Comments to the Author**

1. Is the manuscript technically sound, and do the data support the conclusions?

Reviewer #1: No

Reviewer #2: Yes

Reviewer #3: Yes

Reviewer #4: Yes

2. Has the statistical analysis been performed appropriately and rigorously? 

Reviewer #1: No

Reviewer #2: Yes

Reviewer #3: Yes

Reviewer #4: N/A

3. Have the authors made all data underlying the findings in their manuscript fully available?

Reviewer #1: No

Reviewer #2: Yes

Reviewer #3: Yes

Reviewer #4: Yes

4. Is the manuscript presented in an intelligible fashion and written in standard English?

Reviewer #1: No

Reviewer #2: Yes

Reviewer #3: Yes

Reviewer #4: Yes

5. Review Comments to the Author

Reviewer #1: Short tittle : I suggest to change to “Parechovirus infection diagnosis by RT-LAMP”

Key word: The keywords may be useful to found this work when searched by other researchers. I think that more researchers search by RT-LAMP and not “reverse transcription loop-mediated isothermal amplification”, so I suggest to use this option, but I I leave it to the authors’ consideration.

General : Authors should revise the language and sequence of the text. The authors use many repetitions like .. “Early”, “Early” in sequential sentences, or “HPeV3”…” HPeV3”… “HPeV3”

Abstract :

Authors claim “However, bedside rapid diagnostic laboratory tests for HPeVs, including HPeV3, do not exist.”. I’m not sure that a RT-LAMP can be considered as a “bedside test”, it requires knowledge and time, often not far from an RT-qPCR, and there are several for the diagnosis of these viruses. In fact, RT-LAMP takes 60 minutes, being feasible doing a qPCR in this time using direct PCR.

Abstract Conclusion “This report is the first showing that HPeVs can be detected with RT-LAMP”, This is not a conclusion of this study, theoretically any RNA sequence can be detected by this technique. This sentence should be correct to “This manuscript is the first describing a RT-LAMP for HPeVs diagnosis, allowing a faster, easily and cheaper diagnosis.”

Results : “We created three combinations of primers and confirmed that all primer sets allowed detection of HPeV3 RNAs in only 60 minutes.” This statement can generate some confusions for the readers. 60 minutes could be the duration of any PCR depending specially in the size of amplicon (generally smaller than 200 bp) and the enzymes used, so this sentence do not make sense because posteriorly the authors refer 2 hours (line 68). I understand that 1hour correspond to the PCR time and 2 hours to all the process, but this two times could appear confusing. Remove the statement of the Abstract because the time of PCR is presented in the protocol.

Line 63. Why the authors just refer the nested PCR (more laborious)? Many fast qPCR techniques are described: 10.1128/JCM.01982-12, 10.1128/JCM.00277-08, https://doi.org/10.1016/j.jcv.2016.09.009, https://doi.org/10.1007/s11033-019-05151-5 and many others.

Line 72. Why the authors just used one sequence to design primers. They have previous knowledge of full conservation of the region used?

Line 80. Many times the authors present the sentences with a romantism that is not normally used in scientific papers. Please change the sentence to “HPeV3 viral ribonucleic acid (RNA), used as positive control, was kindly provided by the Health and Food Safety Department at Ishikawa Prefectural Institute of Public Health and Environmental Science (Kanazawa, Ishikawa, Japan).”

Line 83. Present all the Providers as (Company, City, Country)

Line 85 . Use dNTPs instead “deoxyribose nucleoside triphosphates”

Line 85. The methods should not be presented “as a brief”. The objective of material and methods is to allow the full repeatability by other researchers. Please present all the conditions, quantities and other important information.

Line 91 “reverse RT”? What is the R in RT?. The sentence can appear simply as “Complementary DNA (cDNA) was synthesized using Superscript II Reverse Transcriptase…”

Line 93 . I’ve never seen the term “test-positive samples”. Please use, samples tested positive

Line 99. Uniformize all the brands that you use, including minuscles/maiscules. Some times appear “BigDye Terminator cycle sequencing kit”, other “BigDye Terminator Cycle Sequencing Kit”

Line 106/107- Noro / Adeno are not official diminutive for viruses, please remove these references. If you want, you can use HuNoV and HAdV

Line 112 – How the authors measured 50-100ul of stool? It were they consistent with watery diarrhea?

114- Centrifugation should be all stated in G, not rpm, alternatively refer the brand and model of machine.

Line 114 – Present “150 μL”

In all the manuscript, and for example in the line 129, the authors write the manuscript in the first singular person that is not the indicated to scientific papers. “When we attempted to find”… “we could not find” … “Therefore, while shifting … we comprehensively extracted”.

By this reason, I recommend the revision of all the manuscript because it is very difficult to read.

Reviewer #2: The manuscript titled “Rapid molecular diagnosis of Parechovirus infection using reverse transcription loop-mediated isothermal amplification technique” reports a visual isothermal technique for portable detection of parechovirus infection. Overall, I believe the manuscript would be of interest to the community, but I have a number of comments that should be addressed before I can recommend it for publication.

-The phrasing should be improved, as it is difficult to understand what is being stated in a few places.

-There is no limit of detection determined. The authors should address why it was not determined using Probit analysis (their results are presence/absence). They should address this and state this would be future work in the Discussion section.

-The authors also should address why no direct comparison to one step real time RT-PCR was performed, as this would have been the closest assay rival to the RT-LAMP assay they have developed.

-What was the approximate load of viral nucleic acid loaded in reactions for the selectivity work?

-Also, virus names should not be italicized/capitalized when referring to viruses in the manner they are used in the publication; this should only be done when broadly referring to a family, genus, or species.

-Which strains/subtypes of adenovirus and norovirus were used for the selectivity testing.

-Lines 133-138: I am not sure what the authors are referring to here.

-The table listing primers is a little unclear and could be a little better organized with a legend explaining what is being presented.

-The authors list viral nucleic acid in terms of ng/pg. However, in real application genome copies or pfu/ml (or TCID50/ml) is much more useful. What levels of virus do these values approximately translate to? Please state in the manuscript.

-Lines 257-260: This is a bit of an exaggeration, as many viruses can cause illness and this assay only answers if it is one type of virus.

Reviewer #3: The article entitled « Rapid molecular diagnosis of Parechovirus infection using the reverse transcription loop-mediated isothermal amplification technique» is well written and the results are clearly presented. However, some modifications of the manuscript and figures should be done. The authors created three combinations of primers to detect HPeVs using RT-LAMP experiments. They conclude that HPeV infection can be used for the faster diagnosis with less cost.

the paper is accepted for publication.

Reviewer #4: The current study by Yokoyama is nicely written and describes the development of RT-LAMP assay for the HePV. The paper should be accepted for publication in its current form. A minor suggestion is that authors should try to improve the tests to enable it for the detection of different genotypes separately.

6. PLOS authors have the option to publish the peer review history of their article (what does this mean?). If published, this will include your full peer review and any attached files.

---

## [Author Response · Author response to Decision Letter 0]

2 Nov 2021

Emily Chenette,

Editor-in-Chief

PLOS ONE

Dear Professor Emily Chenette:

Thank you for your valuable review. We also wish to thank the reviewers who reviewed our manuscript and provide such valuable suggestions. Here are our responses to the comments from the reviewers.

Journal Requirements:

We carefully revised our manuscript and confirm that it meets PLOS ONE’s style requirements.

2. Thank you for including your ethics statement: "This study was approved by the institutional review board of Graduate School of Medical Sciences at Kanazawa University [Kanazawa, Japan; protocol number: 2014-072(1686)]. This study was conducted according to the Declaration of Helsinki. All experiments were performed in accordance with relevant guidelines and regulations (including informed consent from all participants and parents). We obtained informed consent from the parents when we collected the stools of anonymous febrile infants..”

Please provide additional details regarding participant consent. In the ethics statement in the Methods and online submission information, please ensure that you have specified what type you obtained (for instance, written or verbal, and if verbal, how it was documented and witnessed).

We obtained written informed consent and added this information to the Materials and Methods section (Line 118).

3. Please provide the data or graph for the temperature optimization experiment (line 174–175) as a supplementary file.

Thank you for pointing this out. We added the temperature optimization data (Line 185, S4 Table).

4. Please provide: the name(s) and source(s) of the cultured cells used for virus-infection, as well as a brief description of viral infection methods. If the samples were from a clinical source, please state whether the samples were:

(1) from an established biobank (if so please provide the name and a link)

(2) specifically collected for this study or not

(3) whether the samples were collected through a medically prescribed test

(4) whether the samples were completely de-identified before researchers accessed the samples

The cultured cells are Vero/E6 cells, and the source of the cells is the National Institute of Infectious Diseases in Japan (https://www.niid.go.jp/niid/ja/). The samples were collected through a medically prescribed test and completely de-identified before we accessed the samples. This information was added to the Materials and Methods section (Line 104–107).

5. We note that the grant information you provided in the “Funding Information” and “Financial Disclosure” sections do not match.

When you resubmit, please ensure that you provide the correct grant numbers for the awards you received for your study in the “Funding Information” section.

We revised the funding information as follows:

This project was supported by the Morinaga Foundation for Health & Nutrition (http://www.disclo-koeki.org/02a/00327/index.html). The funders had no role in the conceptualization, analysis, interpretation, or decision to publish this manuscript.

Thank you for your suggestion. The sequence confirmation of the LAMP products and the homology confirmation of the primers have been removed from the phrase “data not shown” because they are not the core parts of our study. Also, the temperature optimization data was added as S4 Table. Please let us know if you need more supplementary data. We will submit it additionally.

Reviewers' comments:

Reviewer #1:

Short title: I suggest to change to “Parechovirus infection diagnosis by RT-LAMP”

Keyword: The keywords may be useful to found this work when searched by other researchers. I think that more researchers search by RT-LAMP and not “reverse transcription loop-mediated isothermal amplification,” so I suggest to use this option, but I leave it to the authors’ consideration.

Thank you for your suggestion. We also found that “RT-LAMP” was more appropriate, so we changed the title as per your suggestion.

General: Authors should revise the language and sequence of the text. The authors use many repetitions like. “Early,” “Early” in sequential sentences, or “HPeV3”…” HPeV3”… “HPeV3”

Regarding the issues you pointed out, the manuscript was proofread by a native English speaker.

Abstract:

Authors claim “However, bedside rapid diagnostic laboratory tests for HPeVs, including HPeV3, do not exist..” I’m not sure that a RT-LAMP can be considered as a “bedside test,” it requires knowledge and time, often not far from an RT-qPCR, and there are several for the diagnosis of these viruses. In fact, RT-LAMP takes 60 minutes, being feasible doing a qPCR in this time using direct PCR.

Thank you for your suggestion. Since our laboratory does not have a qPCR device, we reasoned that the LAMP method, which can detect using only a heat block, would be easier. However, after our discussion, we agree with the reviewer’s opinion. Thus, we changed the sentence to “development of rapid and easier diagnostic laboratory tests for HPeVs is desired” (Line 22).

Abstract Conclusion “This report is the first showing that HPeVs can be detected with RT-LAMP,” This is not a conclusion of this study, theoretically any RNA sequence can be detected by this technique. This sentence should be correct to “This manuscript is the first describing a RT-LAMP for HPeVs diagnosis, allowing a faster, easily and cheaper diagnosis.”

Following your suggestion, the text has been changed (Line 36–38).

Results: “We created three combinations of primers and confirmed that all primer sets allowed detection of HPeV3 RNAs in only 60 minutes.” This statement can generate some confusions for the readers. 60 minutes could be the duration of any PCR depending specially in the size of amplicon (generally smaller than 200 bp) and the enzymes used, so this sentence do not make sense because posteriorly the authors refer 2 hours (line 68). I understand that 1hour correspond to the PCR time and 2 hours to all the process, but this two times could appear confusing. Remove the statement of the Abstract because the time of PCR is presented in the protocol.

As you pointed out, we have deleted “in only 60 minutes” (Line 30).

Line 63. Why the authors just refer the nested PCR (more laborious)? Many fast qPCR techniques are described: 10.1128/JCM.01982-12, 10.1128/JCM.00277-08, https://doi.org/10.1016/j.jcv.2016.09.009, https://doi.org/10.1007/s11033-019-05151-5 and many others.

Because we do not have a real-time PCR machine in our lab, we decided to establish the LAMP method instead of real-time PCR. However, it is difficult to mention that we do not have a real-time PCR machine in the lab.

The second reason is that nested PCR was not originally created to only detect HPeV3 but was established to detect several viruses based on conventional PCR and sequence analysis of amplification products.

The third reason, which overlaps with the first reason, is that we do not have the funds to run qPCR.

Thus, we added the sentences in Line 63 as follows: “The real-time PCR is also widely used. However, it would be useful if there was a faster, easier, and cheaper diagnostic test for HPeVs.”

In the discussion, the following sentences were added:

Line 227: The real-time PCR method is widely used for the current HPeV3 detection.

Line 230: faster, easier, and cheaper

Line 290: In addition, although we believe the LAMP method is useful for screening, it is necessary to confirm whether real-time PCR or the LAMP method is faster and cheaper.

Line 72. Why the authors just used one sequence to design primers. They have previous knowledge of full conservation of the region used?

We also confirmed the sequences of other HPeV3 (for example, GenBank: AJ889918). However, it is already known that this sequence (strain A308-99: AB084913.1) is the most common and almost conserved gene sequence of HPeV3 detected in Japan, so in this study, we used this sequence.

Line 80. Many times the authors present the sentences with a romanticism that is not normally used in scientific papers. Please change the sentence to “HPeV3 viral ribonucleic acid (RNA), used as positive control, was kindly provided by the Health and Food Safety Department at Ishikawa Prefectural Institute of Public Health and Environmental Science (Kanazawa, Ishikawa, Japan).”

As per your suggestion, the text has been changed (Line 80).

Line 83. Present all the Providers as (Company, City, Country)

Thank you for your advice. We added all Providers as (Company, City, Country). We also reconfirmed all Providers.

Line 85.se dNTPs instead “deoxyribose nucleoside triphosphates”

Following your suggestion, the text has been changed (Line 85).

Line 85. The methods should not be presented “as a brief.” The objective of material and methods is to allow the full repeatability by other researchers. Please present all the conditions, quantities and other important information.

It was customarily described as a preposition, so we deleted it.

Line 91 “reverse RT”? What is the R in RT? The sentence can appear simply as “Complementary DNA (cDNA) was synthesized using Superscript II Reverse Transcriptase…”

Thank you very much. This is a simple mistake. We wanted to write “reverse transcriptase” instead of “reverse RT.” The description has been changed as you pointed out.

Line 93.’ve never seen the term “test-positive samples.” Please use, samples tested positive

As per your suggestion, the text has been changed (Line 93).

Line 99. Uniformize all the brands that you use, including minuscles/maiscules. Sometimes appear “BigDye Terminator cycle sequencing kit,” other “BigDye Terminator Cycle Sequencing Kit”

The description has been changed to “BigDye™ Terminator v3.1 Cycle Sequencing Kit.”

Line 106/107- Noro / Adeno are not official diminutive for viruses, please remove these references. If you want, you can use HuNoV and HAdV.

As you suggested, we unified it without using abbreviations. We also modified Figure 2.

Line 112–How the authors measured 50–100ul of stool? It were they consistent with watery diarrhea?

A sample of stool was placed into an Eppendorf tube and lightly centrifuged before measuring its volume. However, when we weighed it, we found that 50–100 μL of the stool we used was equivalent to 75–150 mg, so we changed the description accordingly.

114- Centrifugation should be all stated in G, not rpm, alternatively refer the brand and model of machine.

For the centrifuge we used, 15,000 rpm is equivalent to 20,000g, so we changed the description accordingly.

Line 114–Present “150 μL”

As you suggested, the text has been changed (Line 120).

In all the manuscript, and for example in the line 129, the authors write the manuscript in the first singular person that is not the indicated to scientific papers. “When we attempted to find”… “we could not find” … “Therefore, while shifting … we comprehensively extracted.”

The manuscript was proofread by a native English speaker.

By this reason, I recommend the revision of all the manuscript because it is very difficult to read.

Thank you for these suggestions; we feel that the quality of the revised manuscript is better than the initial version. Thank you for your efforts to produce a better quality treatise. We would be grateful if you could proofread the manuscript once again.

Reviewer #2: The manuscript titled “Rapid molecular diagnosis of Parechovirus infection using reverse transcription loop-mediated isothermal amplification technique” reports a visual isothermal technique for portable detection of Parechovirus infection. Overall, I believe the manuscript would be of interest to the community, but I have a number of comments that should be addressed before I can recommend it for publication.

-The phrasing should be improved, as it is difficult to understand what is being stated in a few places.

We requested a native speaker to proofread the English text once again and confirmed that it was difficult to read.

-There is no limit of detection determined. The authors should address why it was not determined using Probit analysis (their results are presence/absence). They should address this and state this would be future work in the Discussion section.

Thank you for your suggestion. In this study, the analysis was performed using a limited number of cases, so the detection limit was not examined. Therefore, the issue you pointed out is the limitation of this research. We added the sentence “(3) the limit of detection determined using Probit analysis” to Line 288.

-The authors also should address why no direct comparison to one step real-time RT-PCR was performed, as this would have been the closest assay rival to the RT-LAMP assay they have developed.

This suggestion was also made by reviewer #1. Because we do not have a real-time PCR machine in our lab, we decided to establish the LAMP method instead of real-time PCR. However, it is difficult to mention that we do not have a real-time PCR machine in our lab.

Therefore, we added the sentences in Line 63 as follows: “The real-time PCR is also widely used. However, it would be useful if there was a faster, easier, and cheaper diagnostic test for HPeVs.”

In the discussion, the following sentences were added:

Line 227: The real-time PCR method is widely used for the current HPeV3 detection.

Line 230: faster, easier, and cheaper

Line 290: In addition, although we believe the LAMP method is useful for screening, it is necessary to confirm whether real-time PCR or the LAMP method is faster and cheaper.

-What was the approximate load of viral nucleic acid loaded in reactions for the selectivity work?

The amount of viral nucleic acid is listed in the S5 Table. However, for the sake of clarity, we added “applied viral nucleic acid amount is 62–296 ng/sample” to the method section (Line 113).

-Also, virus names should not be italicized/capitalized when referring to viruses in the manner they are used in the publication; this should only be done when broadly referring to a family, genus, or species.

As you pointed out, the italicized virus name has been changed to the official nomenclature.

-Which strains/subtypes of adenovirus and norovirus were used for the selectivity testing.

As for the adenovirus and norovirus samples, we did not confirm the strain/subtypes because the samples rapidly tested positive in the patient’s feces.

“The patients were diagnosed by rapid diagnostic tests.” (Line 111)

In addition, the adenovirus that was distributed is known to be serotype1; therefore, we described it as such (Line 106).

-Lines 133–138: I am not sure what the authors are referring to here.

We have revised these sentences to make them easier to understand.

-The table listing primers is a little unclear and could be a little better organized with a legend explaining what is being presented.

The table has been amended so that the primer information is easy to view for each set.

-The authors list viral nucleic acid in terms of ng/pg. However, in real application genome copies or pfu/ml (or TCID50/ml) is much more useful. What levels of virus do these values approximately translate to? Please state in the manuscript.

“pfu/mL” measures the amount of virus activity. However, our purpose is the amount of virus detected in fecal specimens. In our assay, the viral RNA is extracted directly from the stool (the stool was directly dissolved, and the capsid of the virus was destroyed), so we believe that the virus dies at that point. Thus, the amount of virus activity is not measured. Genome copy is difficult to measure because we cannot perform real-time PCR and do not have a positive control with a known concentration of virus.

-Lines 257–260: This is a bit of an exaggeration, as many viruses can cause illness and this assay only answers if it is one type of virus.

As you pointed out, we found that this sentence was overstated, and we deleted it.

Reviewer #3: The article entitled « Rapid molecular diagnosis of Parechovirus infection using the reverse transcription loop-mediated isothermal amplification technique» is well written and the results are clearly presented. However, some modifications of the manuscript and figures should be done. The authors created three combinations of primers to detect HPeVs using RT-LAMP experiments. They conclude that HPeV infection can be used for the faster diagnosis with less cost. the paper is accepted for publication.

Thank you for your review. On the basis of the suggestions from other reviewers, we modified the manuscript and figures. We are proud that the paper has better quality, but we would appreciate it if you could point out any questions.

Reviewer #4: The current study by Yokoyama is nicely written and describes the development of RT-LAMP assay for the HePV. The paper should be accepted for publication in its current form. A minor suggestion is that authors should try to improve the tests to enable it for the detection of different genotypes separately.

Thank you for your advice. In our study, it was not possible to detect different genotypes separately. This is the limitation of this research. Therefore, in the manuscript (Line 268–270), we described “further experiments are necessary to design HPeV type-specific primers and develop the appropriate LAMP condition.”

We hope that many researchers will be interested in our research and that there will be colleagues who consider working in collaboration with us to establish a rapid diagnostic method for each genotype.

Many thanks.

Sincerely yours,

Tadafumi Yokoyama, M.D., Ph.D.

Department of Pediatrics, Kanazawa University

---

## [Editor Report · Decision Letter 1]

9 Nov 2021

Rapid molecular diagnosis of Parechovirus infection using the reverse transcription loop-mediated isothermal amplification technique

PONE-D-21-06788R1

Dear Dr. Yokoyama,

We’re pleased to inform you that your manuscript has been judged scientifically suitable for publication and will be formally accepted for publication once it meets all outstanding technical requirements.

Kind regards,

Ahmed S. Abdel-Moneim, Ph.D.

Academic Editor

PLOS ONE

---

## [Editor Report · Acceptance letter]

17 Nov 2021

PONE-D-21-06788R1 

Rapid molecular diagnosis of Parechovirus infection using the reverse transcription loop-mediated isothermal amplification technique 

Dear Dr. Yokoyama:

I'm pleased to inform you that your manuscript has been deemed suitable for publication in PLOS ONE. Congratulations! Your manuscript is now with our production department. 

Kind regards, 

on behalf of

Prof. Ahmed S. Abdel-Moneim 

Academic Editor

PLOS ONE